# Grapevine and Wine Metabolomics-Based Guidelines for FAIR Data and Metadata Management

**DOI:** 10.3390/metabo11110757

**Published:** 2021-11-03

**Authors:** Stefania Savoi, Panagiotis Arapitsas, Éric Duchêne, Maria Nikolantonaki, Ignacio Ontañón, Silvia Carlin, Florian Schwander, Régis D. Gougeon, António César Silva Ferreira, Georgios Theodoridis, Reinhard Töpfer, Urska Vrhovsek, Anne-Francoise Adam-Blondon, Mario Pezzotti, Fulvio Mattivi

**Affiliations:** 1UMR AGAP, Montpellier University, CIRAD, INRAE, Institut Agro-Montpellier SupAgro, 34060 Montpellier, France; savoi.stefania@gmail.com; 2Department of Food Quality and Nutrition, Edmund Mach Foundation, Research and Innovation Centre, Via Edmund Mach 1, 38010 San Michele all’Adige, Italy; silvia.carlin@fmach.it (S.C.); urska.vrhovsek@fmach.it (U.V.); fulvio.mattivi@unitn.it (F.M.); 3SVQV, University of Strasbourg, INRAE, F-68000 Colmar, France; eric.duchene@inrae.fr; 4UMR PAM Université de Bourgogne/Agro Sup Dijon, Institut Universitaire de la Vigne et du Vin, Jules Guyot, F-21000 Dijon, France; maria.nikolantonaki@u-bourgogne.fr (M.N.); regis.gougeon@u-bourgogne.fr (R.D.G.); 5Laboratorio de Análisis Del Aroma y Enología, Departamento de Química Analítica, Facultad de Ciencias, Instituto Agroalimentario de Aragón (IA2), Universidad de Zaragoza, Calle de Pedro Cerbuna 12, 50009 Zaragoza, Spain; ionta@unizar.es; 6Julius Kühn-Institute, Institute for Grapevine Breeding Geilweilerhof, D-76833 Siebeldingen, Germany; florian.schwander@julius-kuehn.de (F.S.); reinhard.toepfer@julius-kuehn.de (R.T.); 7CBQF—Centro de Biotecnologia e Química Fina.—Laboratório Associado, Escola Superior de Biotecnologia, Universidade Católica Portuguesa, Rua de Diogo Botelho, 1327, 4169-005 Porto, Portugal; asferreira@porto.ucp.pt; 8Laboratory of Analytical Chemistry, Department of Chemistry, Aristotle University of Thessaloniki, 54124 Thessaloniki, Greece; gtheodor@chem.auth.gr; 9Université Paris-Saclay, INRAE, URGI, 78026 Versailles, France; anne-francoise.adam-blondon@inrae.fr; 10Department of Biotechnology, University of Verona, 37134 Verona, Italy; 11Department of Cellular, Computational and Integrative Biology, CIBIO, University of Trento, 38123 Trento, Italy

**Keywords:** open-data, plants, omics, *Vitis*, wine, metabolites, chromatography, mass spectrometry

## Abstract

In the era of big and omics data, good organization, management, and description of experimental data are crucial for achieving high-quality datasets. This, in turn, is essential for the export of robust results, to publish reliable papers, make data more easily available, and unlock the huge potential of data reuse. Lately, more and more journals now require authors to share data and metadata according to the FAIR (Findable, Accessible, Interoperable, Reusable) principles. This work aims to provide a step-by-step guideline for the FAIR data and metadata management specific to grapevine and wine science. In detail, the guidelines include recommendations for the organization of data and metadata regarding (i) meaningful information on experimental design and phenotyping, (ii) sample collection, (iii) sample preparation, (iv) chemotype analysis, (v) data analysis (vi) metabolite annotation, and (vii) basic ontologies. We hope that these guidelines will be helpful for the grapevine and wine metabolomics community and that it will benefit from the true potential of data usage in creating new knowledge being revealed.

## 1. Introduction

Thanks to the increasing availability of thousands of sequenced plant genomes [1], and the parallel uses of high-throughput analyses of next-generation sequencing techniques, such as the popular RNA sequencing [2], profiling the entire plant transcriptome and performing studies on gene expression during development and/or in response to biotic and environmental conditions [3] is now quite straightforward. In the past few years, the advancement of the methodologies based on liquid and gas chromatography (LC and GC) coupled with mass spectrometry (MS) and nuclear magnetic resonance (NMR) spectrometry opened a new field of research, metabolomics [4,5,6], that makes it possible to perform large-scale measurements of hundreds or even thousands of metabolites in one run with targeted or untargeted approaches [7]. However, due to the chemical complexity of the metabolome, which differs from the four nucleotides with similar chemical properties that characterize the transcriptome, it is currently not possible to profile the entire plant metabolome using a single extraction protocol and a single analytical technique. The metabolomic space covered by an untargeted approach method can vary according to the analytical system, from dozens of major compounds in a NMR experiment [8,9] to several hundreds or thousands of compounds for HRGC-MS [10] or HPLC-MS experiments [11]. Moreover, ultrahigh-resolution mass spectrometry (Fourier-transform ion cyclotron resonance mass spectrometry, FT-ICR-MS) [12,13] makes it possible to record thousands of signals for metabolic fingerprinting. Targeted approach methodologies usually cover a few to several dozen known metabolites [14]. Very often, therefore, several different protocols need to be integrated to achieve the desired coverage of the metabolome within a single study [15,16]. Moreover, the data are presented in a myriad of formats. Data from targeted experiments are quantitative and can be expressed as milliequivalents of multiple reference standards, with reference to the fresh or dry weight, etc. The data from untargeted experiments are the results of a process where the analyzed metabolome is not defined a priori, and this includes several unknown metabolites. Different sample preparation, instrumental analysis, and data analysis protocols will deliver complementary (but not conflicting) datasets and, therefore, possibly slightly different conclusions. This higher complexity requires highly organized data and metadata management, and metabolomic data must be combined with a detailed set of metadata to be correctly read and reused outside the original experiment.

With the recent developments at the level of LC-MS and GC-MS instrumentation (faster, more sensitive, more accurate, and with higher resolution) and data analysis bioinformatics tools, the number of metabolomics applications in the field of grapevine and wine research is increasing exponentially. Many researchers from the grape and wine science have started to work with big omics and delivered, over the last two decades, a huge volume of interesting data and new knowledge. However, in contrast to the grapevine transcriptomic field, where uploading the raw data in public repositories is mandatory for publication, in metabolomics, this action is just recommended and is very rarely mandatory [17].

Nevertheless, submitting the raw data to ad-hoc public repositories has become necessary in order to manage this exponential increase in biological data and their related metadata. This is further confirmed by the complexity and the variability of the protocols in use by the different laboratories. Often experimental designs, sample preparation, and analytical protocols are only partially described, and the terminology or ontologies used are not harmonized. In 2007, Sumner and colleagues [18] set up the first general guidelines to describe a metabolomic experiment, and since then, several groups have improved these general rules in various specific fields, especially in biofluid metabolomes (e.g., [19,20]). However, there are not many specific guidelines for grapevine and wine metabolomics research, and in that respect, the aim of the Cost Action CA17111 INTEGRAPE consortium was to fill this gap.

Nowadays, a researcher has multiple public repositories to upload metabolite analysis raw files to. A few are specific to the field of metabolomics (e.g., MetaboLights and Metabolomics Workbench), whereas others collect data from all the fields (e.g., figshare). MetaboLights is an open-access, curated database for metabolomics experiments connected with their raw data and associated metadata [21,22]. The database is cross-species, cross-technique, and covers metabolite structures and their reference spectra as well as their biological roles, locations, and concentrations. It is part of the ELIXIR infrastructure [23], hosted by the European Bioinformatics Institute (EMBL-EBI), and is one of the recommended metabolomics repositories for several leading journals in the field. The Metabolomics Workbench [24] is another repository for metabolomics data and metadata, as well as metabolite standards, protocols with tutorials, training, and analysis tools. The availability of both data and metadata is of the utmost importance in order to be able to adhere to the FAIR principles [25] that make data, experiments, and results Findable, Accessible, Interoperable, and Reusable.

The goal of this work is to provide all researchers within the grapevine and wine science fields with specific guidelines to implement best practices, improve the quality, availability, and usefulness of the data and the associated metadata, and facilitate the indexing of the produced dataset. A second purpose is encouraging the grapevine and wine science community to follow these guidelines as much as possible and, finally, to harmonize the data and metadata resources into a common, shared, and user-friendly format.

## 2. Methods

Each study created on public repositories has a unique alphanumeric study ID (e.g., MTBLS000 or ST0000). The study ID cannot be modified, and it is needed when referencing the study in manuscripts or elsewhere, together with the relevant URL. Please note that the study ID can be used to retrieve the dataset in MetaboLights, but as it is only mentioned in the materials and method section, it is not sufficient to obtain the indexing of the dataset in the Data Citation Index (DCI) of Clarivate Web of Science, which would be desirable in order to connect any published paper to the FAIR data present in the public repository. In order to facilitate the automatic inclusion in the DCI, it is recommended to specifically insert this persistent, resolvable identifier in a citation in the manuscript when the data are published. This also means that the data must be uploaded well in advance of the submission of the manuscript.

A general overview of the key protocols for the organization of data and metadata, especially when the final goal is to upload such information and raw files in public repositories (e.g., MetaboLights) under the FAIR principles, is shown in Figure 1. In addition, detailed information is presented in the following paragraphs.

Study descriptors are the fingerprints of the study, and they may include information such as the authors’ names and IDs, an abstract giving a brief overview, publication references, and a list of keywords for indexing purposes. One of the most critical sections representing metadata enabling reproducibility is a series of protocols used in the study. Usually, these include a sample collection and sample preparation protocol (Table 1 and Table 2). Moreover, one or multiple protocols describing the analyses performed are necessary. An example is a protocol for liquid or gas chromatography, followed by a second for mass spectrometry (Table 3). Other protocols are related to the steps taken for data transformation and, lastly, a protocol containing how the metabolites were annotated and quantified (Table 4). Finally, to adhere to the FAIR principle, the raw data, possibly in an open format, needs to be uploaded and linked to a study with the aim that other researchers can access it.

### 2.1. Sample Collection

Within the protocol adopted in this study, it is essential to describe the origin of the samples (source, organism, genus, species, intraspecific name, organism part, etc.), to include some technical description of the experiments (row and plant spacing, rootstock, planting date, training system, soil management technique, etc.), any relevant treatments, biotic or abiotic stresses; time points or cultural operations, such as pruning, hedging, fertilization, pesticide spraying, etc., and the collection and storage procedure (Table 1). Moreover, information on how to describe experiments involving wine samples is reported. We highly recommend the use of OIV (International Organization of Vine and Wine) terminologies, ontologies, and standards. Other necessary metadata on the experimental design are the developmental stage, phenological description, geographic location, agronomical practices, etc. A set of recommendations on adequately describing a grapevine experiment to guarantee interoperability and standard ontologies between different datasets is based on the international metadata standard for Plant Phenotyping experiments (MIAPPE, www.miappe.org, accessed on 30 October 2021) [26]. The MIAPPE checklist is a pdf file that lists all the items recommended to describe an experiment, from the general characteristics of an investigation/study to the biological samples proposing descriptors for experimental factors and environmental variables.

### 2.2. Extraction (Sample Preparation)

This protocol is intended to describe any extraction or sample preparation method applied to the sample before analysis (Table 2). It is very important to report information on any quality control samples (QC) prepared for the assay, e.g., pooled samples, blanks, standards mixtures, etc. QCs are used to handle any analytical variability, for example, by monitoring the performances of the instrument, any shifts in compounds retention times, signal intensity, to reduce noise, to correct for batch effect or systematic errors, and they should be injected at the beginning of the experiment and then repeatedly every 5 to 10 samples, depending on the size of the sample set. The pooled QC samples should represent the overall composition of all the samples, encompassing all the major developmental stages or treatments of the samples under analysis, and they should be prepared and extracted in the same way as the other samples. Blanks are another type of QC, which should be prepared using the same protocol of the study, except that the sample matrix is to be replaced by nothing, if it is solid, or by the extraction solvent (or water), if it is liquid. Blanks will help find any artifacts, contamination, or unwanted by-products. One or more internal standards can be added to the real samples, blanks, and pooled samples, which can be helpful during the normalization process. Randomization of the sample extraction/sample preparation and injection order is also encouraged to help in reducing bias in the interpretation of the results. In particular, a systematic scheme for the sample run order should be created by combining the experimental design with complete or group-wise randomization [19]. Complete randomization is the preferred solution, but partial randomization is also acceptable in case of a large number of samples to be analyzed.

### 2.3. Chromatography and Mass Spectrometry

By definition, chromatography is an analytical, separative technique where compounds are eluted in different retention times, based on molecular characteristics and interaction type, that uses ion exchange, surface adsorption, partition, and size exclusion mechanisms. For reasons of reproducibility, it is necessary to provide details of the instrument and column(s), mobile phase and gradient, and settings, such as temperatures, flow rate, injection volume, etc. (Table 3). Moreover, the chemical methods ontology (https://www.ebi.ac.uk/ols/ontologies/chmo, accessed on 30 October 2021) can help find vocabularies to describe the methods and tools used to collect data in chemical experiments, such as chromatography and mass spectrometry.

A mass spectrometer measures the abundance of the mass-to-charge ratio (*m*/*z*) of the ions generated by the ionization process of the eluted metabolites, previously separated by GC or LC. Within this protocol, it is mandatory to provide details of the instrument used, ion source, ionization mode (positive/negative), *m*/*z* range, and specific parameters such as temperatures, voltages, flow rates, scan rates, etc. (Table 3). Since the analyzed metabolites are not predefined in metabolomics, method validation is rather tricky. However, a minimum reporting of instrumental performance parameters is encouraged. For example, the description of the nature and method(s) used is essential to ensure instrumental sensitivity, selective, linearity, stability, resolution, and mass accuracy. The QC samples distribution in a PCA plot is a good indicator.

These guidelines can be applied even in the case of less extensive experimental settings and methodologies, such as targeted chromatography or spectroscopy (e.g., FTIR Fourier-transform infrared spectroscopy) and beyond (e.g., genomic approaches), especially for sample collection, classification, preparation, and data processing, which are supposed to be reported with the same level of detail.

### 2.4. Data Transformation and Metabolite Identification

The data acquired by the instrument need to be converted and processed before any further data and statistical analysis. In order to obtain a raw data matrix, features (*m*/*z* intensity related to a specific retention time) need to be extracted, often with an automatic pipeline or thanks to dedicated commercial software. Hence, this protocol requires the details of the methods, pipeline, or software used to transform the raw data and all the steps and parameters used to extract the features (Table 4). More helpful information can also be found in [19], Appendix 4.

Metabolite identification is one of the most critical phases in metabolomics, requiring expertise, and is a time-consuming task. In the case of a targeted method, features can be identified by means of reference standards or library matching and quantified by their calibration curve. The identification of unknown features extracted from untargeted methods is more challenging, although the availability of several databases might help in this task [19,20]. In Table 4, we describe the three main methods used for metabolite annotation confidence level. It is mandatory to use one of them in reporting the results because the use of common names is often confusing.

## 3. Results

To the best of our knowledge, up to September 2021, we were able to retrieve fourteen publicly available grapevine metabolome datasets [15,16,32,33,34,35,36,37,38,39] and seven others related to wine science [40,41,42,43,44,45,46], with the majority of such studies deposited in the repository MetaboLights. It is foreseeable that the number will quickly increase in the near future as the application of metabolomics to plant science increases. These guidelines are intended to assist the grapevine and wine metabolomics community in publishing their data in compliance with the FAIR protocol.

In MTBLS2876 [32,47], there are data and metadata on the most important classes of grapevine leaf metabolites (primary compounds, lipids, phenols, and volatile organic compounds) detected at several time points after artificial inoculation of the oomycete *Plasmopara viticola* in resistant and susceptible grape varieties.

MTBLS784 [33,48] reported a study on the pre-harvest application of methyl jasmonate on *Vitis labrusca* grapes to verify the interaction with the phenolic compounds content by UPLC-HRMS-QTOF.

MTBLS968 [34,49] is a study about the accumulation profiles of terpene metabolites in three Muscat table grape cultivars through HS-SPME-GCMS.

The following three datasets (MTBLS898, MTBLS982, MTBLS984) [16,50,51,52] comprehensively characterized the metabolic response of Merlot grape berries exposed to water deficit at different developmental stages by UPLC-MS/MS, SPME-GC-MS, and UPLC-DAD assays for the analysis of phenolic, volatiles and carotenoids compounds.

The MTBLS392 study [35,53] aimed to explore the core microbiota and metabolome of *Vitis vinifera* L. cv. Corvina grapes and musts during different vintages, by two-dimensional gas chromatography time-of-flight mass spectrometry.

Studies MTBLS897, MTBLS892, and MTBLS889 [15,54,55,56] represent the datasets of broad research investigating the impact of water deficit on the secondary metabolism of white grapes using large scale metabolites analyses in a season characterized by prolonged drought.

The study with ID MTBLS346 [36,57] investigated the downstream metabolic changes of Silcora and Thompson seedless grape cultivars when genetically modified through the insertion of the DefH9-iaaM gene into their genome. The effect of the genetic modification upon the grape metabolome was evaluated by ^1^H-NMR and exploratory data analysis and chemometrics methods.

MTBLS209 [37,58] is a comparative study of the metabolome of the three grape berry tissues (skin, pulp, seeds) of three American *Vitis* species (*Vitis cinerea*, *Vitis californica*, *Vitis arizonica*) together with four interspecific hybrids, and seven *Vitis vinifera* cultivars.

The authors of the MTBLS85 study [38,59] presented an automated data analysis of high throughput high-performance liquid chromatography with diode array detection (HPLC-DAD) data using multivariate curve resolution; a case study on the stability of isoprenoids in grape extracts under two different experimental regimes serves to illustrate the potential of the method.

And, finally, study MTBLS39 [39,60] described the plasticity of the berry metabolome of the cultivar Corvina sampled during development at different geographical locations.

Concerning wine metabolomics, study MTBLS1677 [40,61] analyzed a set of 917 wines of Czech origin using nuclear magnetic resonance spectroscopy with the aim to build and evaluate multivariate statistical models and machine learning methods for the classification of such wines based on type, variety, and location.

The aim of the next project, with the raw data and metadata deposited with ID MTBLS1443 [41,62], was to register the metabolome of 11 single-cultivar single-vintage Italian red wines from 12 regions across Italy, each one produced in their terroirs under ad hoc legal frameworks in a LC-MS-based untargeted single-batch analysis, to guarantee their quality and origin.

In MTBLS2330 [42,63], the authors aimed to study the impact of water deficit on the concentration of key flavor and phenolic secondary metabolites of both red and white wines, linking the previously observed drought-induced grape compositional changes to the wines.

Study MTBLS212 [43,64] untangled the wine metabolome by combining untargeted SMPE-GCxGC-TOF-MS and sensory analysis to profile Sauvignon blanc co-fermented with seven different yeasts.

The MTBLS137 dataset [44,65] reported the UPLC-QTof MS untargeted analysis of *Vitis vinifera* L. leaves, collected in Italy and Germany from two fungus-resistant grape varieties (PIWI), Regent and Phoenix. It is an illustrative example of the use of the MetaDB pipeline. MetaDB was developed in order to combine with a user-friendly web-based, different bioinformatic tool used in metabolomics, which takes care of (a) metadata organization, (b) creation of randomized sequences including QC sample, (c) data quality evaluation, (d) data storage organization, (e) data analysis, and (f) submission to public repositories.

MTBLS55 [45,66] is a study about the influence of storage condition and duration on the “chemical age” of red wines.

And finally, study ST000006 [46,67], deposited at Metabolomics Workbench repository, reported the data of a GC-TOF analysis on seventeen white wines for investigating the chemical basis for mouthfeel properties in wine.

## 4. Discussion

Although limited in number, the above-described public datasets cover a big part of the technological and scientific themes of the grapevine and wine community. Their most common aim was to investigate the cultivar/geographical differentiation, and second, to study the abiotic and biotic stresses. However, considering the scientific literature, there are still many fields that are widely present in the literature but are not covered by these datasets, from viticulture practices (rootstock, training systems, soil, leaf removal, etc.) to winemaking techniques (enzymes/microorganism/coadjuvant, physical technologies, special wines, etc.) and shelf life (packaging, exposure to oxygen, light, temperature, and the evolution during storage, etc.). It is positive to see that the above-described datasets retrieved from public repositories were produced using an array of diverse, modern, or traditional instrumental/analytical techniques, such as gas/liquid chromatography, various mass spectrometers, nuclear magnetic resonance, and diode array.

Engaging in completely open science may be unrealistic and even undesirable for many researchers. However, the use of FAIR repositories is becoming mandatory by many funding agencies. We urge researchers working in viti-oenology to learn and adopt existing, robust tools that are immediately helpful and important to them in order to improve the visibility, quotability, and preservation of their data, which will be safely archived and ready to reuse.

And we all know, and it is especially true in the era of big data, that reproducibility and transparency are key tenets of the scientific process that requires reliability and generality of results and methods [68]. Of course, making a dataset publicly available is not enough since data and metadata management according to the FAIR principles is equally important. This holistic process could help make the studies easier to find for humans and computer tools (ontologies, IDs, rich and indexed metadata, etc.), accessible to all (open, free, and universally applicable protocols), interoperable with other (meta)data and datasets, and optimized to be reused and useful to other scientists. The connection between metabolomics and the other omics (genomics, transcriptomics, and proteomics) towards multi-omics, which for the moment is still in its infancy, will be easier and more straightforward. Research institutes without direct access to multi-scientific platforms will be less penalized and limited because they might have access to FAIR datasets and can build directly on primary work. The real potential of big data, which is impossible to explore in one study, will be unlocked. The reproducibility and transparency of the published knowledge will be improved due to the curation of the coded workflows (shared in public repositories) and revision (journal’s peer revision). Metabolite annotation, which is now the bottleneck of most studies, will be substantially improved by systematically using metabolite IDs and reference spectra sharing and via access to existing, searchable databases for confirmation of the chemical features of interest in different experimental contexts.

On the other hand, we acknowledge that this process is often complex and time-consuming and that guidelines, tutorials, and training are necessary. Since the general guidelines are often complicated and difficult to understand, it is necessary to support the research community and not work in isolation. COST CA17111 INTEGRAPE decided to write the first specific guidelines for grapevine and wine metabolomics FAIR data and metadata management.

It is encouraging to discover that the great majority of the publicly available metabolomic datasets in grapevine and wine science, and especially the studies uploaded to MetaboLights (Appendix A), are filed under FAIR principles, even though some were released before the FAIR publication. They all include the appropriate protocols, the use of ontologies and IDs, and have raw files uploaded on the website. There is only one study on a different public repository, which lacks many important metadata parameters and does not include raw files (but an excel document), but this is one of the oldest ones since it was released in 2013. The long experience and the curation protocol of MetaboLights make this website an excellent hub for the metabolomics community. However, a better connection between MetaboLights, VIVC IDs, and MIAPPE ontologies is strongly suggested.

## 5. Concluding Remarks

In the era of molecular system biology and integrative studies beyond single omics techniques, the availability of raw data supported with a rich set of metadata is now compulsory in all omics research. To advance in such specific fields, the research community requires data-sharing, common language, open science, and straight collaborations between the different scientific disciplines. It is time that the «metabolomics» scientists learn from the successful experience of their «genomics» brothers and start to manage, organize and share their data and metadata according to the FAIR principles.

Within this frame, the COST Action CA 17111 INTEGRAPE was launched to integrate data at different levels to maximize the power of omics in order to improve reproducibility, transparency, and accessibility. It does this by optimizing the data flows and interoperability between datasets, different analytical tools, data standards, and ontologies and by providing recommendations and tools with the final purpose of grapevine improvement. Therefore, we strongly promote these phenotypic and metabolomics/chemotypic guidelines to our grape and wine community, knowing that the most valuable experiment is the one already filed and made available on a dedicated repository online.

## Figures and Tables

**Figure 1 metabolites-11-00757-f001:**
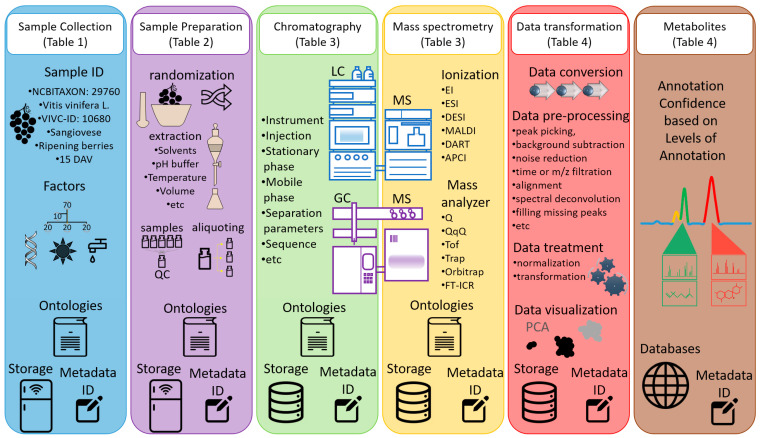
Overview of the required data and metadata management steps supplied with basic information on grapevine phenotyping and chromatography mass spectrometry techniques. Detailed information is included in each table.

**Table 1 metabolites-11-00757-t001:** Sampling protocol (plant materials or wine sample).

Field	Description
Source	Where the samples were collected. The use of an ID is recommended (https://ror.org/, accessed on 30 October 2021). Example: Fondazione Edmund Mach collection (ID 0381bab64), or experimental winery, winery, supermarket, etc.
Organism	An identifier for the organism at the species level. The use of the NCBI taxon ID is recommended. For *Vitis vinifera* the ID is 29760. (https://www.ncbi.nlm.nih.gov/taxonomy/, accessed on 30 October 2021).
Specie(s)	According to the standard scientific nomenclature, species name (formally: specific epithet) for the organism under study (e.g., *Vitis vinifera* L.).
Intraspecific name(s)	Three field codes might be necessary to identify the exact plant material used in an experiment. Field 1: code for the institution. Please refer to WIEWS codes from the FAO (http://www.fao.org/wiews/en/, accessed on 30 October 2021) or ROR codes (https://ror.org, accessed on 30 October 2021) for research organizations. Field 2: type of plant material. The most commonly used denomination for grapevine material is the variety name. We recommend using a standard name, such as the “prime name” extracted from the VIVC database (http://www.vivc.de, accessed on 30 October 2021). The type of plant material can be classified with (i) the five-digit VIVC code for identified varieties, (ii) “PRO” for genotypes from bi-parental crosses, (iii) “TL” for transgenic lines, (iv) “ESL” for lines regenerated from anthers or somatic tissues, or (v) nothing when the type of plant material is not characterized. Field 3: code used to identify the accession available in the institute. For plants from genetic resources, the unique accession number of the EU-Vitis Database (http://www.eu-vitis.de/, accessed on 30 October 2021) is recommended. Examples: FRA038_VIVC10077_274Col49 for Riesling clone number 49 available at INRAE Colmar. FRA038_PRO_41207Col0011E for a genotype in the progeny from a cross between Riesling and Gewürztraminer. DEU098-1980-315 for a specific Riesling accession in the *Vitis* collection of JKI Geilweilerhof.
Organism part	A reliable description of biological samples requires a shared vocabulary for the organ collected. The grapevine ontology anatomy is available at http://agroportal.lirmm.fr/ontologies/GAO (accessed on 30 October 2021) or https://data.inrae.fr/dataset.xhtml?persistentId=doi:10.15454/SBXYSV (accessed on 30 October 2021).
Developmental stages	Several scales to describe the grapevine developmental stages are available [27,28,29] and can be used in a grapevine experiment. Here we propose to add some accuracy to the descriptions of these stages [30]. **Dates for the main development stages**. A bud is counted as “broken” if a green (or red) tip is visible (BBCH 07, Baggiolini C). The budbreak date is determined by interpolation between several successive records, as the day when 50% of the buds left after pruning have reached this stage. For flowering (BBCH 65, Baggiolini I), the flowering date is determined as the day when 50% of the flower caps detach or fall. For véraison (BBCH 85, Baggiolini M), the most relevant definition is “softening” and not “color change”, in order to record values that can be compared between white and colored genotypes. The date of véraison is determined as the day when 50% of the berries are soft. A reliable estimation of the percentage of soft berries should be based on touching at least 100 berries (20 on five plants, for example). **Phenological descriptors for the berries.** Four types of berry samples can be distinguished: (i) green berries, (ii) ripening berries, (iii) mix of green and ripening berries, (iv) harvested berries. In order to allow comparisons between experiments, we propose to provide the following data to best characterize a sample, ranked by decreasing relevance. For green berries: (i) number of days after flowering (DAF) or before véraison (as defined above), (ii) heat sums calculated with the degree days (usually above 10 °C, otherwise to be specified), starting at flowering, (iii) single berry weight or volume. For ripening berries: (i) number of days after véraison (DAV) (as defined above), (ii) heat sums (usually base 10 °C, otherwise to be specified) after véraison, (iii) single berry weight or volume, (iv) sugar concentration, (v) acidity parameters (pH, titratable acidity, malic acid concentration, tartaric acid concentration, potassium concentration). For harvested berries (post ripening berries, BBCH 99): (i) number of days after harvest. **Phenological descriptors for the leaves**: (i) age (number of leaves above, when the apex is active), (ii) position (from the base of the shoot). Deviations due to the needs of experimental settings are to be explained in detail.
Tissue harvesting method	Register the details about how the sampling occurred in the field/vineyard. For example, report if the samples were directly frozen and how (e.g., liquid N_2_, dry ice, freeze clamping, etc.), the date and time of collection, the place of collection, if samples were washed to remove unwanted external components (e.g., soil), shipping time and temperature, and sample storage before further preparation (e.g., −80 °C for two weeks).
Harvest protocol	Include information about the harvest date and period, if it was made manually or mechanically, the time of the day (morning, afternoon, night), grape sanitary status, crop yields, crushing and pressing devices and settings, yield of must or wine, pre-fermentative processing (e.g., grape cooling, sulfitation, etc.), information related to the experiment, etc.
Sample Type (Wine)	Describe at which point in the production line the samples were collected (must, day of fermentation, end of alcoholic fermentation, end of malolactic fermentation, after barrel aging, etc.).
Winemaking protocol	Include information about additions (additives and processing aids, amount, timing, method of mixing), microorganism (source, genus, species, commercial name), must volumes, container type and volume, maceration technique, temperature, length of fermentation, use of oak/wood barrels (or wood staves or chips), micro-oxygenation, first or second fermentation, disciplinary if PDO, and other information related to the experiment. The following two documents from OIV can help: https://www.oiv.int/public/medias/4954/oiv-oeno-567a-2016-en.pdf (accessed on 30 October 2021). https://www.oiv.int/public/medias/7713/en-oiv-code-2021.pdf (accessed on 30 October 2021).
Basic oenological analysis	Include all available measurements (e.g., soluble solids, pH, titratable acidity, fermentable nitrogen, sugars, ethanol, residual sugar, free and total sulfites, malic and lactic acid, total phenols, absorbance at 420 and 520 nm, volatile acidity, Cu, Fe) and other information relevant for the experiment. The next three documents from OIV can help: https://www.oiv.int/en/technical-standards-and-documents/methods-of-analysis/compendium-of-international-methods-of-analysis-of-wines-and-musts (accessed on 30 October 2021). https://www.oiv.int/en/technical-standards-and-documents/methods-of-analysis/the-guidelines-in-oenology (accessed on 30 October 2021). https://www.oiv.int/en/technical-standards-and-documents (accessed on 30 October 2021).
Commercial wine	If the study is based on commercial wines sourced from the market, provide information about source (supermarket, wine shop, winery, etc.), date of sampling, price, PDO data, cultivar(s), winery, region, country, vintage, color, basic oenological analysis, lot number, bottling date, and other information related to the experiment.
Packaging	Include information about packaging material (glass bottle, tetra-pak, metallic, plastic, etc.), packaging transparency (flint, green, amber, dark), volume, stopper (corks–natural, colmated, agglomerated, 1 + 1, micro-agglomerated-, synthetic, screw caps, glass, etc.), and other information related to the experiment.
Replicate sampling and analyses	The number of samples represents the “sample size”, and all the samples together represent the “sample set”. A part of the “sample set”, thus a “sample subset”, can be used for the analysis. A minimum of triplicate (*n* = 3) biological sampling is proposed, with *n* = 5 preferred. It is more critical to measure biological replicates than technical ones.
Storage conditions and aliquoting prior to extraction	Report information about temperature, duration, possible additives, volumes, container, and atmosphere. Samples can be divided into small aliquots. Report method, volume or weight, and number of aliquots.
Relocation or shipping info	Provide information about the shipping conditions (if relevant).
Sample IDs list	Sample ID list connected to the above information.

**Table 2 metabolites-11-00757-t002:** Extraction (sample preparation) protocol.

Field	Description
Randomization	Report if the sample preparation order was randomized and how (https://www.random.org/sequences/, accessed on 30 October 2021).
Extraction parameters	Solvent(s), pH and ionic strength of the buffer, solvent temperature and volume(s) per quantity of tissue, internal standard(s), number of replicate extracts (technical and biological replicates), sequential extraction, and extraction time.
Concentration/Dilution	Extract concentration, dilution, and resolubilization processes (e.g., dried under nitrogen, solubilized in methanol).
Enrichment	Extract enrichment (e.g., solid-phase extraction, desalting, molecular cut-off, ion exchanges, rotary vapor).
Extract treatments	Extract cleanup and/or use of additives (e.g., ultrafiltration, centrifugation, the addition of antioxidants, pH change).
Derivatization	Report the protocol of derivatization (the chemical used, temperature, time, etc.).
Quality Control Sample(s)	Report if a QC pooled sample was prepared using extracts of the entire “sample set” or a “sample subset”. In addition, report the method (volume or weight from each sample and total amount of the QC pooled sample).
Reference Material	Report if any biological reference material and/or a standard mixture was used and how it was purchased or prepared. This material can also be used as QC samples.
Blanks	Report how the blank sample was prepared.
Aliquoting	Aliquots prepared during or after the sample preparation (code, volume, number). This includes the QC samples.
Storage–Relocation	Extract storage (e.g., temperature, duration, atmosphere, volumes, containers, etc.) and/or relocation (e.g., temperature, duration, atmosphere, places).
Internal standard(s) addition	Internal standard(s) at any stage(s).
Samples ID list	Update the Sample ID list, including the names or the IDs of the extracts. Often more than one extraction protocol is applied to the same samples.

**Table 3 metabolites-11-00757-t003:** Chromatography and Mass spectrometry protocol.

Field	Description
Instrument	Manufacturer, model number, software package and version. The majority of the instruments can be found in the EMBL/EBI ontology (https://www.ebi.ac.uk/ols/ontologies/ms, accessed on 30 October 2021). If this is the case, we recommend the use of the ontologies; if not, use free text.
Injection	Auto-injector (manufacturer, model, type, software, injector/loop volume, wash cycles, solvents, volume, SPME parameters, automatic derivatization, injector temperature, split or splitless mode, and ratio, etc.).
Stationary phase	Separation column(s) and pre/guard column (manufacturer, model/name, stationary phase composition, particles, internal diameter, physical parameters, length, parameters of 2D chromatography, etc.).
Mobile phase	Mobile phase (e.g., gases, solvents, buffers, pH) including their preparation protocol (information of the type of flasks, pipette, degasser, etc.) and post-column modifiers (if applied).
Separation	Separation parameters (sample temperature, mobile phases composition(s), gradient profile, column temperature, flow rate(s), pressure, etc.).
Sequence	Sequence duration and length of stay of the sample in the sampler before analysis. Report if the “sample set” or “sample subset” order was randomized and the frequency of the QC analysis (all types of QC samples used).
Sample introduction and delivery	Direct infusion (continuous or not) after GC, CE, or LC separation.
Ionization source	Ionization mode (EI, APCI, ESI, etc.), polarity (positive or negative), vacuum pressure, skimmer/focusing lens voltages (e.g., capillary voltage, etc.), gas flows (e.g., nebulization gas, cone gas, source temperature, etc.).
Mass analyzer	Type of analyzer (e.g., quadrupole, ion-trap, time-of-flight, FT-ICR, including combinations of these for hybrid instruments). The majority of the analyzers can be found in EMBL/EBI ontology (https://www.ebi.ac.uk/ols/ontologies/ms, accessed on 30 October 2021).
Acquisition mode and parameters	For a single quadrupole instrument, the scan modes are full scan and sim; for a triple quadrupole instrument, common modes are full scan, product scan, precursor scan, neutral loss scan and MRM. In high-resolution MS (QTof and Orbitrap), common scan modes are: (a) full scan; (b) data-dependent acquisition, such as MS/MS; and c) data-independent acquisition, such as Swath, Sonar, MSall, MSn, MSe, MSc2, AIF-MS2, vDIA, bbCID. All the parameters of the acquisition mode should be reported, such as the m/z scan range, polarity(ies), scan speed, collision energy(ies), cycle time, resolution, mass accuracy, and spectral acquisition rate, vacuum pressure, various voltages, etc.
Ion Mobility	Type (DTIMS, TIMS, DMS, etc.), place (e.g., before or after the quadrupole), buffer gas, separation parameters.
Technique-specific sample preparation	Re-suspension of sample (e.g., in MeOH:water 1:1 with 0.2% formic acid), derivatization, volume injected, and internal calibrant(s) added (if relevant).
Calibration	Calibration compound(s) and mode.
Lock spray	Concentration, lock mass, flow rate, and frequency.
Analysis or Assay ID	Update the Sample ID list, including the names or the IDs of the raw assay files. This name could include the date of the analysis, the order of the analysis, and information about the protocol. The sample with the ID: SAN12 could have as Assay ID: 20201214_055_SAN12_RP_NEG, where the first part gives information about the date of analysis, the second the order of the analysis, the third the sample ID, and the fourth about the chromatography protocol.

**Table 4 metabolites-11-00757-t004:** Data transformation/conversion parameters protocol and metabolite identification.

Field	Description
Raw data format	Report the format of the original raw data, as registered by the instrument and its software.
Data conversion	Often the raw data are converted to “open” (or not) formats, such as net.CDF, XML, MZml, etc., for their further analysis. Report the software and its version used for the data conversion and the parameters used.
Data pre-processing	The original or the converted data are often processed before the statistical analysis. For the MS data, the process might include peak picking, background subtraction, noise reduction, time or *m*/*z* filtration, alignment, spectral deconvolution, smoothing, binning, data reduction, filling missing peaks, etc. Report the software and its version used together with the parameters. The most popular software are MZmine, XCMS, MSdial, metaMS, Progenesis QI, and MetAlign.
Data treatment	The obtained peak table from the data pre-treatment can be further treated with normalization and scaling tools. First, report the software, its version, and the parameters used. Then, inspecting the data for drift correction or outliers’ detection is envisaged.
Annotation confidence	The correct peak or metabolite annotation is crucial for the interpretation of the results, and it is important to provide information as far as the confidence of each annotation will allow by applying one of the below-listed annotation level protocols.
Four levels annotation [18]	This is the most common method used to report the annotation confidence in metabolomics. It includes the following levels of annotation: **1. Identified compounds.** A minimum of two independent and orthogonal data relative to an authentic compound analyzed under identical experimental conditions is proposed as necessary to validate non-novel metabolite identifications (e.g., retention time/index and mass spectrum, retention time and NMR spectrum, accurate mass and tandem MS, accurate mass and isotope pattern, full ^1^H and/or ^13^C NMR, 2-D NMR spectra). **2. Putatively annotated compounds.** This level is applied when the annotation is made without chemical reference standards, based upon physicochemical properties and/or spectral similarity with public/commercial spectral libraries or literature. If spectral matching is utilized in the identification process, then the authentic spectra used for the spectral matching should be described appropriately or libraries made publicly available. **3. Putatively characterized compound classes.** The annotation is based upon characteristic physicochemical properties of a chemical class of compounds or by spectral similarity to known compounds of a chemical class (e.g., hexose, carotenoid, lipid, anthocyanin, etc.). **4. Unknown compounds.** Although unidentified or unclassified, these metabolites can still be differentiated and quantified based upon spectral data.
Five levels annotation [31]	This is the second most used method to report the annotation confidence in metabolomics. It includes the following levels of annotation: **Level 1**: Confirmed structure represents the ideal situation, where the proposed structure has been confirmed via appropriate measurement of a reference standard with MS, MS/MS and retention time matching. If possible, an orthogonal method should also be used. **Level 2**: Probable structure indicates that it was possible to propose an exact structure using different evidence. For Level 2a: a library that involves matching literature or library spectrum data where the spectrum–structure match is unambiguous. Care is needed when comparing spectra recorded with different acquisition parameters (e.g., resolution, collision energy, ionization, MS level, retention behavior) to ensure the validity of the match; decision criteria should be clearly presented. For Level 2b: diagnostic represents the case where no other structure fits the experimental information, but no standard or literature information is available for confirmation. Evidence can include diagnostic MS/MS fragments and/or ionization behavior, parent compound information, and the experimental context. **Level 3**: Tentative candidate(s) describes/e a “grey zone”, where evidence exists for possible structure(s), but the information for one exact structure only is insufficient (e.g., positional isomers). **Level 4**: Unequivocal molecular formula is possible when a formula can be unambiguously assigned using the spectral information (e.g., adduct, isotope, and/or fragment information), but insufficient evidence exists to propose possible structures. The MS/MS could be uninformative, contain interferences, or not even exist. **Level 5**: Exact mass (*m*/*z*) can be measured in a sample and be of specific interest for the investigation but lack information to assign even a formula. Screening and nontarget methods allow the tracing of these masses in other investigations, but level 5 indicates that no unequivocal information about the structure or formula exists. It is even possible to record the MS/MS of a level 5 mass and save it as an “unknown” spectrum in a database. This level should only apply to a few masses of specific interest since it would be counterproductive to label all masses in a sample as level 5. Blank measurements should be used to ensure the substance does not arise from sample preparation or measurement.
Metabolomics Society’s Metabolite Identification Task Group	The metabolomics community recently released a new method by introducing subclasses (A-F) for unambiguous metabolite annotation. https://drive.google.com/file/d/1PJLdPCkz8ymX8SgZ4Wl5Sw4ZG-dlyWWU/view, accessed on 30 October 2021. The proposed levels are: **A: Known enantiomer**. A single defined enantiomer or a single defined achiral metabolite. Molecular formula, structure, and stereochemistry, including chirality, are known. Usually requires isolation of metabolite and complete structure determination or chiral chromatography on metabolite in a mixture to prove chirality and matching of two orthogonal pieces of data with an authentic chemical standard. For achiral metabolites, it requires the matching of two orthogonal pieces of data with authentic chemical standards (e.g., RT and MS/MS mass spectrum). **B: Known diastereomer**. One of two enantiomers. Known molecular formula, structure, and stereochemistry but unknown chirality. Requires matching of two orthogonal pieces of data with authentic chemical standards (e.g., RT and MS/MS mass spectrum). **C: Known structure/DB position.** One of a number of stereoisomers, e.g., E/Z geometric or *cis*-/*trans*-ring isomers. Known molecular formula and structure but unknown stereochemistry. Requires matching of two orthogonal pieces of data with authentic chemical standards (e.g., RT and MS/MS mass spectrum). **D: Known functional group**. One of a number of positional isomers. Known molecular formula and metabolite class but unknown structure, e.g., high-resolution mass spectrometry provides unique and unambiguous single molecular formula, and additional data proves metabolite class membership. **E: Known formula**. One of a number of possible compounds of known molecular formula. Known molecular formula but unknown structure, e.g., high-resolution mass spectrometry provides the unique and unambiguous single molecular formula. **F: Known structural class**. Specific spectral features defining a structural class. Unknown molecular formula but a known class of metabolite; characteristic signals of metabolite class in the sample. **G: Known formula**. Specific spectral futures. Unknown molecular formula; characteristic signals of unknown metabolite in the sample.
Metabolite ID	Identify the annotated metabolites with a unique identifier (ID) corresponding to one of the following databases: Chemical entities of biological interest (ChEBI), HMDB, FoodDB, KEGG, Chemspider, PubChem, COlleCtion of Open Natural ProdUcTs ID, or CAS.

## Data Availability

Not applicable.

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
