# Peer review of "Grapevine and Wine Metabolomics-Based Guidelines for FAIR Data and Metadata Management"

_metabolites, 2021, doi:10.3390/metabo11110757_

Round 1
Reviewer 1 Report
This technical note addresses the issue of adopting a common set of reporting standards within the grapevine and wine sciences, which is currently an ongoing topic within the metabolomic and lipidomic fields. The goals and reporting standards seem logical, and appear to address most metadata that would be useful in defining a metabolomics dataset, ranging from the samples themselves and their subsequent collection, the extraction of compounds and subsequent analysis on the platform of choice, followed finally by the data analysis pipeline.
However, there are two main points that need addressing, firstly, the grammar needs to be refined, as this will greatly improve the readability of the article, and I’ve addressed these below (see line numbers).
Secondly, the results and discussion section could do with refining, as it is currently a list of all the wine and grapevine research studies found within open access repositories. I would suggest extracting the salient points from these studies and write them in a more discussive format. As this is a technical note, I can see why this was done (as a “results” section), however, as the main aim of this article is to address the FAIR data management principles, I would suggest at least some discussion as to whether the studies described actually adhere to these standards, and to what extent (to what extent are these guidelines already being followed within the field?).
It might also be worth mentioning what, in general terms, is currently being explored within the wine and grapevine community (from the studies presented here); for example, differentiation of cultivars, abiotic/biotic stress, modelling and classification (this would link into the previous point of de-listing the results and discussion section). Also, as mentioned in the introduction, as it is envisaged that metabolomics will be more widely adopted within the field (hence the establishment of these guidelines), what do you see as some of the points/issues metabolomics might help address within the field?
Specific points:
Table 2. Extraction parameters: “number of replicate extracts”: might be worth breaking into technical and biological extractions (I believe this point is made earlier in the text).
Table 3. Acquisition mode and parameters: b) data dependant acquisition: I believe the strict definition of DDA is a MS1 survey scan followed by n number of MS/MS scans, informed by the survey scan, taking into account inclusion/exclusion lists for example. I think SIM and MRM would fall under a targeted approach.
Line 31: “make the data publicly available easier”, easier to write “make the data more easily available” or similar.
Line 32: “more journals require to share data and metadata”, “more journals now require authors to share data and metadata” or similar.
Line 39: “unhiding”, probably better to use “revealing” or “unlocking”.
Line 48: “in the past ‘few’ years”
Line 50: “together with the nuclear magnetic”, remove “the”.
Line 51: I don’t think it’s “so-called”, I believe it is actually called that; “opened a new field of research, metabolomics”.
Line 57: “the metabolomic space coverage with” reads better if “the metabolomic space covered by an untargeted approach can vary”.
Line 58: “from many dozen major compounds of an NMR experiment” reads better if “from dozens of major compounds in an NMR experiment” or similar.
Line 60: “and is typical of thousands of signals of signals”, I think this paragraph should be split for clarity, as I’m not sure if this next sentence relates to the previous statement, and is quite a long sentence.
Line 62: “The targeted approach methodologies usually cover from a few to several dozens of known metabolites”, would read better if “The targeted approach methodologies usually cover a few to several dozen known metabolites”.
Line 67: “experiments are results of” should be “experiments are the result of”.
Line 77/78: “from the grape and wine science started to work”, should be “from the grape and wine sciences have started to work”.
Line 83/84: “relative metadata information”, could be subjective, but I would remove information, as metadata is inherently information, and improves sentence flow.
Line 90: “especially in biofluids metabolome” should be “especially in biofluid metabolomes” or just “biofluids”.
Line 95: This sentence could be made clearer “A few are specific to the field of metabolomics, whereas others collect data from all the fields”, or something similar.
Line 103: I would amend to “The Metabolomics Workbench is another repository for metabolomics data and metadata, as well as metabolite standards, protocols with tutorials, training and analysis tools.”
Line 108: I would remove “to” and amend to “the goal of this work is to provide all researchers within the grapevine and wine science field” or similar.
Line 110: “handiness”, I think you might mean usefulness, but I would change the word if this is incorrect.
Line 111: “Further purpose is encouraging”, I would rephrase as this isn’t very clear.
Line 120: “report” should be “reporting”. Again, “information” could be removed, but subjective.
Line 122: “quantidied” to “quantified”.
Line 128: “With the term “study”, this sentence needs rephrasing, something like “The term “study” is intended to encompass the entire study, however it can be further subdivided”.
Line 172: “Another type of QCs” should be “another type of QC”.
Line 175-177: I think part of the sentence is repeated, so sounds like it should be “One or more internal standards can be added to the real samples, blanks and pooled samples, which can be helpful during the normalisation process”.
Table 3: “Direct infusion (continue or not)”, should be “continuous” I think.
Line 201: “settings”, I think it might be “experimental settings and methodologies” as you also refer to FTIR.
Line 203: “these guidelines can be applied. Especially…”, I think a comma is warranted “these guidelines can be applied, especially for sample collection, classification… and is supposed to be reported with the same level of detail”, or something similar.
Line 212: “Transform” spelling.
Line 215: “requiring-expertise, and is a time consuming task”. Needs to be amended.
Line 221: “the use of the common names”, either “the use of the common name” or “the use of common names”, and “often” spelling.
Table 4: “as far as the confidence of each annotation will allow”? Also, “the annotation of all ‘features’ is impossible” or “all of them”.
Author Response
Dear reviewer,
We would like to thank you for your valuable comments and suggestions. We took them all in consideration. Please find below our answers.
This technical note addresses the issue of adopting a common set of reporting standards within the grapevine and wine sciences, which is currently an ongoing topic within the metabolomic and lipidomic fields. The goals and reporting standards seem logical, and appear to address most metadata that would be useful in defining a metabolomics dataset, ranging from the samples themselves and their subsequent collection, the extraction of compounds and subsequent analysis on the platform of choice, followed finally by the data analysis pipeline.
Thanks for the positive comments
However, there are two main points that need addressing, firstly, the grammar needs to be refined, as this will greatly improve the readability of the article, and I’ve addressed these below (see line numbers).
We apologise for the grammatical errors. We applied all suggested changes. Moreover, the final document was proofread by a native English speaker service.
Secondly, the results and discussion section could do with refining, as it is currently a list of all the wine and grapevine research studies found within open access repositories. I would suggest extracting the salient points from these studies and write them in a more discussive format. As this is a technical note, I can see why this was done (as a “results” section), however, as the main aim of this article is to address the FAIR data management principles, I would suggest at least some discussion as to whether the studies described actually adhere to these standards, and to what extent (to what extent are these guidelines already being followed within the field?).
All the above suggestion are correct. Initially the idea was to make a short manuscript, but the reviewer has right. The Discussion was poor. According to both reviewer’s suggestions we added a new Discussion section and therefore, we discussed the studies founded in public repositories in a more discursive way. We added also a new supplementary Table (S1) where these studies were listed with informative column regarding the FAIR information indicating to what extent the metabolome datasets presented are in agreement with their guidelines.
It might also be worth mentioning what, in general terms, is currently being explored within the wine and grapevine community (from the studies presented here); for example, differentiation of cultivars, abiotic/biotic stress, modelling and classification (this would link into the previous point of de-listing the results and discussion section). Also, as mentioned in the introduction, as it is envisaged that metabolomics will be more widely adopted within the field (hence the establishment of these guidelines), what do you see as some of the points/issues metabolomics might help address within the field?
Thanks again for the suggestion. Now you can find in the Discussion section, information about the thematic explored by these studies, and the important thematic that are missing. Moreover, we included a paragraph about how these guidelines could positively influence the grapevine and wine science.
Specific points:
Table 2. Extraction parameters: “number of replicate extracts”: might be worth breaking into technical and biological extractions (I believe this point is made earlier in the text).
Done
Table 3. Acquisition mode and parameters: b) data dependant acquisition: I believe the strict definition of DDA is a MS1 survey scan followed by n number of MS/MS scans, informed by the survey scan, taking into account inclusion/exclusion lists for example. I think SIM and MRM would fall under a targeted approach.
The reviewer is correct, the phrase might be confusing. We deleted SIM and MRM from the DDA.
Line 31: “make the data publicly available easier”, easier to write “make the data more easily available” or similar.
Done
Line 32: “more journals require to share data and metadata”, “more journals now require authors to share data and metadata” or similar.
Done
Line 39: “unhiding”, probably better to use “revealing” or “unlocking”.
Done
Line 48: “in the past ‘few’ years”
Done
Line 50: “together with the nuclear magnetic”, remove “the”.
Done
Line 51: I don’t think it’s “so-called”, I believe it is actually called that; “opened a new field of research, metabolomics”.
Done
Line 57: “the metabolomic space coverage with” reads better if “the metabolomic space covered by an untargeted approach can vary”.
Done
Line 58: “from many dozen major compounds of an NMR experiment” reads better if “from dozens of major compounds in an NMR experiment” or similar.
Done
Line 60: “and is typical of thousands of signals of signals”, I think this paragraph should be split for clarity, as I’m not sure if this next sentence relates to the previous statement, and is quite a long sentence.
Done
Line 62: “The targeted approach methodologies usually cover from a few to several dozens of known metabolites”, would read better if “The targeted approach methodologies usually cover a few to several dozen known metabolites”.
Done
Line 67: “experiments are results of” should be “experiments are the result of”.
Done
Line 77/78: “from the grape and wine science started to work”, should be “from the grape and wine sciences have started to work”.
Done
Line 83/84: “relative metadata information”, could be subjective, but I would remove information, as metadata is inherently information, and improves sentence flow.
Done
Line 90: “especially in biofluids metabolome” should be “especially in biofluid metabolomes” or just “biofluids”.
Done
Line 95: This sentence could be made clearer “A few are specific to the field of metabolomics, whereas others collect data from all the fields”, or something similar.
Done
Line 103: I would amend to “The Metabolomics Workbench is another repository for metabolomics data and metadata, as well as metabolite standards, protocols with tutorials, training and analysis tools.”
Done
Line 108: I would remove “to” and amend to “the goal of this work is to provide all researchers within the grapevine and wine science field” or similar.
Done
Line 110: “handiness”, I think you might mean usefulness, but I would change the word if this is incorrect.
Done
Line 111: “Further purpose is encouraging”, I would rephrase as this isn’t very clear.
Done
Line 120: “report” should be “reporting”. Again, “information” could be removed, but subjective.
Done
Line 122: “quantidied” to “quantified”.
Done
Line 128: “With the term “study”, this sentence needs rephrasing, something like “The term “study” is intended to encompass the entire study, however it can be further subdivided”.
Done
Line 172: “Another type of QCs” should be “another type of QC”.
Done
Line 175-177: I think part of the sentence is repeated, so sounds like it should be “One or more internal standards can be added to the real samples, blanks and pooled samples, which can be helpful during the normalisation process”.
Done
Table 3: “Direct infusion (continue or not)”, should be “continuous” I think.
Done
Line 201: “settings”, I think it might be “experimental settings and methodologies” as you also refer to FTIR.
Done
Line 203: “these guidelines can be applied. Especially…”, I think a comma is warranted “these guidelines can be applied, especially for sample collection, classification… and is supposed to be reported with the same level of detail”, or something similar.
Done
Line 212: “Transform” spelling.
Done
Line 215: “requiring-expertise, and is a time consuming task”. Needs to be amended.
Done
Line 221: “the use of the common names”, either “the use of the common name” or “the use of common names”, and “often” spelling.
Done
Table 4: “as far as the confidence of each annotation will allow”? Also, “the annotation of all ‘features’ is impossible” or “all of them”.
Done
Best Regards,
Dr. Panagiotis Arapitsas
Reviewer 2 Report
This manuscript proposes important guidelines for the metabolomics community.
There are just a few important points and a few minor points that it would be good to correct.
Minor points :
L68 : "not previously defined" should be "not defined a priori"
L83 : "their relative metadata" should be "their related metadata"
L175-177 : It is not clear what is meant by the following sentence : "One or more 175 internal standards can be added to the real samples, blanks and pooled samples can be added one to several internal standards, and can helpful during the the normalisation process."
L179 : It would be better to replace "is also requested" by "is also encouraged"
L182-183 : replace "where metabolites" by "where compounds"
L202 : Replace "e.g. FTIR (Fourier-transform infrared spectroscopy)" by "e.g. MIR (Mid infrared spectroscopy)"
Table 4 : "binning" not "bining"
L226 : "and other seven" should be "and another seven"
L234 : "in resistance and susceptible grape varieties" should be "in resistant and susceptible grape varieties"
L293 : "which takes care" should be "which takes care of"
L306-307 : remove the 2 "will" from the sentence.
Important points :
L214 : Verify "in [22 - Appendix 4]". I could not find any appendix in reference 22.
Table 4 : Verify "https://drive.google.com/file/d/1PJLdPCkz8ymX8SgZ4Wl5Sw4ZG-dlyWWU/view)." I could not access it.
In "Results and Discussion", the authors should indicate to what extent the metabolome datasets they present are in agreement with their guidelines.
Author Response
Response to reviewer
Dear reviewer,
We would like to thank you for your valuable comments and suggestions. We took them all in consideration. Please find below our answers.
This manuscript proposes important guidelines for the metabolomics community.
Thanks for the positive comments.
There are just a few important points and a few minor points that it would be good to correct.
Minor points:
L68 : "not previously defined" should be "not defined a priori"
Done
L83 : "their relative metadata" should be "their related metadata"
Done
L175-177 : It is not clear what is meant by the following sentence : "One or more 175 internal standards can be added to the real samples, blanks and pooled samples can be added one to several internal standards, and can helpful during the the normalisation process."
Done
L179 : It would be better to replace "is also requested" by "is also encouraged"
Done
L182-183 : replace "where metabolites" by "where compounds"
Done
L202 : Replace "e.g. FTIR (Fourier-transform infrared spectroscopy)" by "e.g. MIR (Mid infrared spectroscopy)"
Regarding a specific request of changing FTIT with MIR, we prefer not to follow the recommendation to set with MIR the focus on the spectroscopic wavelength range. Instead, we would prefer to focus on the broadly used methodology FTIR. Of course, the MIR spectra is the most important spectral range in FTIR. Nevertheless, the widely used wine analysers scan a wavelength range of 926-5012 cm-1 (e.g. Data for Foss Winescan SO2) that covers as well NIR (700-2500) as MIR (2500 – 50000). The OIV Resolution OIV/OENO 390/2010 (https://www.oiv.int/public/medias/1239/oiv-oeno-390-2010-en.pdf ) supports this assessment where MIR is used as a subtopic (2.5) of FTIR (2).
Table 4 : "binning" not "bining"
Done
L226 : "and other seven" should be "and another seven"
Done
L234 : "in resistance and susceptible grape varieties" should be "in resistant and susceptible grape varieties"
Done
L293 : "which takes care" should be "which takes care of"
Done
L306-307 : remove the 2 "will" from the sentence.
Done
Important points :
L214 : Verify "in [22 - Appendix 4]". I could not find any appendix in reference 22.
Table 4 : Verify "https://drive.google.com/file/d/1PJLdPCkz8ymX8SgZ4Wl5Sw4ZG-dlyWWU/view)." I could not access it.
We corrected the reference, thanks for finding the mistake. The correct reference is now 19 Appendix 4. We also verify the link, and it is possible to access it. It is a pdf file shared by the Metabolite Identification Task Group of the Metabolomics Society. We notice that, when copying the link, sometimes the last parenthesis was copied too, creating problems in accessing the file. We apology for the inconvenience, and we remove such parentheses from the text.
In "Results and Discussion", the authors should indicate to what extent the metabolome datasets they present are in agreement with their guidelines.
Thanks for the suggestion. According to the two reviewer’s suggestions we added a new Discussion section and therefore, we discussed the studies founded in public repositories in a more discursive way. We added also a new supplementary table (S1) where these studies were listed with informative column regarding the FAIR information indicating to what extent the metabolome datasets presented are in agreement with their guidelines.
Moreover, the final document was proofread by a native English speaker service.
Best Regards,
Dr. Panagiotis Arapitsas
Reviewer 3 Report
The Technical Note "" is a good example of an informative article that allows, when reading it, to understand how wine production is studied at a metabolic level. Figure 1 is especially attractive. However, the tables are too dense. Table 1 could gain value if it were lightened. Table 2 would be much more instructive if it were transformed into a figure, similar to figure 1. The same I think regarding table 3. Table 4 can also be simplified for a better understanding.Author Response
Response to reviewer
Dear reviewer,
We would like to thank you for your valuable comments and suggestions. We took them all in consideration. Please find below our answers.
The Technical Note "" is a good example of an informative article that allows, when reading it, to understand how wine production is studied at a metabolic level. Figure 1 is especially attractive. However, the tables are too dense. Table 1 could gain value if it were lightened. Table 2 would be much more instructive if it were transformed into a figure, similar to figure 1. The same I think regarding table 3. Table 4 can also be simplified for a better understanding.
We thank the reviewer for the positive comments. We shorten all the tables by eliminating some rows and unnecessary wording sentences.
Figure 1 is a summary of all four tables, that can be used as a graphical abstract to quickly visualized the study. Therefore, we did not create new figures as the main information of Table 2 were already present in Figure 1.
Best Regards,
Dr. Panagiotis Arapitsas